# Human attention-guided visual perception is governed by rhythmic oscillations and aperiodic timescales

Isabel Raposo[1,2], Ian C. Fiebelkorn[3], Jack J. Lin[4,5], Josef Parvizi[6], Sabine Kastner[7,8], Robert T. Knight[9], Assaf Breska[10], Randolph F. Helfrich[1,11]*

1 Hertie-Institute for Clinical Brain Research, University Medical Center Tübingen, Tübingen, Germany, 2 International Max Planck Research School for the Mechanisms of Mental Function and Dysfunction, University of Tübingen, Tübingen, Germany, 3 University of Rochester Medical Center, Rochester, New York, United States of America, 4 Department of Neurology, UC Davis, Sacramento, California, United States of America, 5 Center for Mind and Brain, UC Davis, Davis, California, United States of America, 6 Department of Neurology and Neurological Sciences, Stanford University, California, United States of America, 7 Princeton Neuroscience Institute, Princeton, New Jersey, United States of America, 8 Department of Psychology, Princeton University, Princeton, New Jersey, United States of America, 9 Department of Psychology and the Helen Wills Neuroscience Institute, UC Berkeley, California, United States of America, 10 Max Planck Institute for Biological Cybernetics, Tübingen, Germany, 11 Departments of Psychology and Neurology and the Wu Tsai Institute, Yale University, New Haven, Connecticut, United States of America

* randolph.helfrich@gmail.com

## Abstract

Attention samples visual space sequentially to enhance behaviorally relevant sensory representations. While traditionally conceptualized as a static continuous spotlight, contemporary models of attention highlight its discrete nature. But which neural mechanisms govern the temporally precise allocation of attention? Periodic brain activity as exemplified by neuronal oscillations as well as aperiodic temporal structure in the form of intrinsic neural timescales have been proposed to orchestrate the attentional sampling process in space and time. However, both mechanisms have been largely studied in isolation. To date, it remains unclear whether periodic and aperiodic temporal structure reflect distinct neural mechanisms. Here, we combined computational simulations with a multimodal approach encompassing five experiments, and three different variants of classic spatial attention paradigms, to differentiate aperiodic from oscillatory-based sampling. Converging evidence across behavior as well as scalp and intracranial electroencephalography (EEG) revealed that periodic and aperiodic temporal regularities can theoretically and experimentally be distinguished. Our results extend the rhythmic sampling framework of attention by demonstrating that aperiodic neural timescales predict behavior in a spatially-, context-, and demand-dependent manner. Aperiodic timescales increased from sensory to association cortex, decreased during sensory processing or action execution, and were prolonged with increasing behavioral demands. These results reveal that multiple, concurrent temporal regularities govern attentional sampling.

**Data availability statement:** All relevant data are within the paper and have been made publicly available in the Supporting information files. Source code is available through Zenodo: 10.5281/zenodo.15404714.

**Funding:** This work was funded by the Baden Wuerttemberg Foundation (Postdoc Fellowship to RFH; https://www.bwstiftung. de/de/), German Research Foundation, Emmy Noether Program (DFG HE8329/2-1 to RFH; https://www.dfg.de/en/research-funding/ funding-opportunities/programmes/individual/ emmy-noether), Hertie Foundation, Network for Excellence in Clinical Neuroscience (to RFH; https://www.ghst.de/en/studying-the-brain/ creating-structures/hertie-network-of-excel- lence-in-clinical-neuroscience), US National Institute of Neurological Disorders and Stroke (2R01NS021135 to RTK; R01NS078396 and R21NS113024 to JP; https://www.ninds.nih. gov/), US National Institute of Mental Health (1POMH109429 to RTK; 1R01MH137624 and 1P50MH132642 to SK; R01MH109954 and P50MH109429 to JP; https://www.nimh. nih.gov/), US National Science Foundation (BCS1358907 and BCS1850938 to JP; https:// www.nsf.gov/). The funders had roles in conceptualization of the study design, data col- lection and curation, supervision, and editing of the manuscript.

**Competing interests:** The authors have declared that no competing interests exist.

**Abbreviations:** $BF_{10}$, Bayes factors; ECoG, elec- trocorticography; EEG, electroencephalography; HFB, high-frequency band; PFC, prefrontal cortex; RTs, reaction times; sEEG, stereotacti- cally placed depth electrodes.

## Introduction

Attention selects and amplifies behaviorally relevant sensory information to mitigate the effects of the brain's inherent limited processing capacity [1,2]. Traditionally, attention has been conceptualized as a 'static spotlight' that boosts the representa- tion of a selected stimulus until the next relevant item is selected [3]. Contemporary theories suggest that attention operates as a 'blinking spotlight', where multiple behaviorally relevant items or spatial locations are sequentially sampled [4,5]. Sev- eral lines of research converged on the notion that neural theta-band activity (~3–8 Hz) might constitute a viable neural signature of the sequential sampling process [4–11]. Neural oscillations reflect wide-spread neural phenomena that are charac- terized by distinct phases of high and low neural excitability giving rise to discrete windows of heightened or diminished perceptual sensitivity. However, other studies indicated that it remains challenging to causally link rhythmic electroencephalography (EEG) activity to time-varying behavioral performance. Specifically, the commonly employed analyses do not necessarily test for the presence of rhythmic activity, but quantify the presence of temporal structure, which is not necessarily periodic (rhyth- mic or oscillatory) [12]. Thus, it had been suggested that several previous reports might conflate periodic and aperiodic temporal regularities. To date, there is little consensus how periodic and aperiodic temporal structure should be quantified and to which extent they might be related [13,14].

In addition to studies exploring the rhythmic nature of attention, another line of inquiry has investigated the functional relevance of *intrinsic neural timescales* [15–18]. Intrinsic neural timescales are often conceptualized as 'temporal integration windows' over which specific sensory information can be maintained and processed [15]. Thus, timescales become progressively longer along the cortical hierarchy. The shortest timescales are typically observed in sensory areas, while the longest inte- gration windows are evident in association areas, such as the prefrontal cortex (PFC) [19,20]. Timescales are quantified as the time constant of the exponential decay function of the neural autocorrelation function and the aperiodic temporal structure of intrinsic timescales reflects a key organizing principle of the hierarchical network organization in the brain [21–23].

A study recording single-unit activity from area V4 in non-human primates during spatial attention demonstrated that timescales are dynamically adjusted according to task demands [24]. Neuronal timescales are longer when the selected stimulus is within the neuron's receptive field, as there is temporal integration of information and sustained activity. Non-human primate and human intracranial EEG studies demon- strated that timescales are functionally dynamic and become progressively longer when information is actively maintained in working memory—especially in PFC [16,17]. Here, we addressed whether similar principles apply to human spatial atten- tion. Specifically, we assessed whether aperiodic temporal structure governs human spatial attention at the level of time-resolved behavior, assessed with both scalp and intracranial EEG. We examined intrinsic timescales and the link to rhythmic brain activity in five independent studies based on three well-established spatial attention

paradigms. Moreover, we introduced a simulation approach to demonstrate that periodic and aperiodic temporal regularities can theoretically and experimentally be disentangled.

## Results

Periodic (or oscillatory) neural activity can be characterized as distinct "bumps" that rise above the $1/f$ decay function of the electrophysiological power spectrum, while aperiodic temporal structure can be estimated from the time constant (or timescale) $\tau$ from the exponential decay of the autocorrelation function. We first employed a simulation approach to demonstrate that periodic and aperiodic temporal structure may provide complementary (non-redundant) information.

### Separation of periodic and aperiodic temporal regularities

To assess whether timescales vary as a function of well-established spectral parameters, such as oscillatory frequency, amplitude, or $1/f$ noise characteristics, we first simulated two extreme scenarios, a purely sinusoidal signal and a purely aperiodic signal. We simulated various oscillations (ranging from 1 to 20 Hz) with varying amplitudes and extracted the timescale from the autocorrelation function. In this physiologically implausible scenario, we observed that timescales scaled inversely and non-linearly (exponential decay) with the underlying peak frequency ($rho = -1$, $p < 0.0001$; Fig 1A), while oscillatory peak amplitude exhibited no statistically significant correlation with moderate evidence in favor of the null hypothesis ($rho = 0.05$, $p = 0.7097$, Bayes factors [$BF_{10}$] = 0.176).

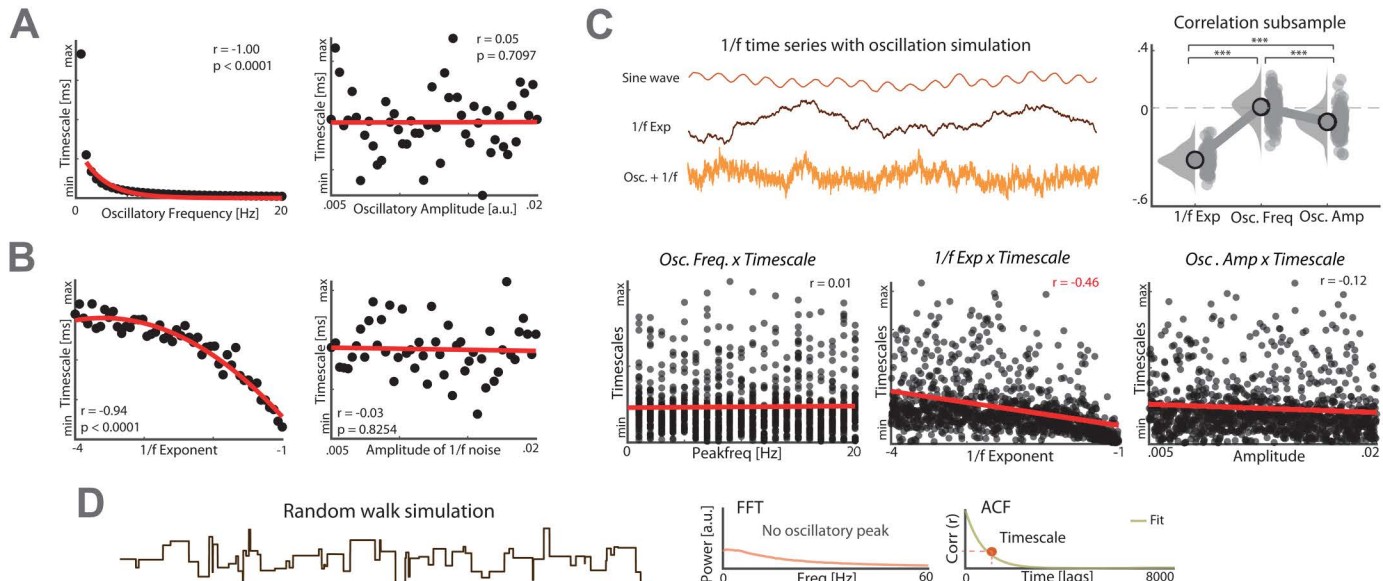

**Fig 1. Analytic distinction of periodic and aperiodic temporal structures in simulations. (A)** Simulation of a sine wave with different frequency and amplitude peaks. Left: Strong correlation detected between timescales and peak frequency. Right: No correlation observed between timescales and peak amplitude. **(B)** Simulation of aperiodic $1/f$ noise with variables exponents and amplitude. Left: Timescales and $1/f$ exponent were strongly correlated. Right: No correlation observed between $1/f$ noise amplitude and timescales. **(C)** Simulation of several timeseries that exhibit both periodic and aperiodic temporal structures with different $1/f$ exponents, frequencies, and amplitudes. Upper left: No statistically significant correlation was observed between timescales and peak frequency of 1,000 timeseries with different exponents and oscillations ($r = 0.01$, $p = 0.8305$, $BF_{10} = 0.045$). Upper right: Strong significant correlation was identified between timescales and $1/f$ noise ($r = -0.46$, $p < 0.0001$). Lower left: A moderate correlation between timescales and peak amplitude was observed ($r = -0.12$, $p = 0.0001$). Lower right: Bootstrapped and subsample correlation coefficients (100 iterations) reveal a significant difference between the parameters amplitude, frequency and $1/f$ exponent ($F = 406.74$, $p < 0.0001$). **(D)** Simulation of random walk signal that contains no periodic activity patterns (lower left), but exhibits a characteristic timescale that can be inferred from the autocorrelation function (lower right; red for represents timescale). Abbreviations: Freq: Frequency. Amp: Amplitude. Exp: Exponent. PSD: Power spectral density. FFT: Fast Fourier Transform. ACF: Autocorrelation Function. The individual values for panel C are included in S1 Data.

Secondly, we assessed another extreme scenario, i.e., a signal that exhibited no oscillations, but solely 1/$f$ activity with varying physiologically plausible exponents (ranging from 1 to 4). These simulations indicated that the aperiodic times-cales correlated strongly with the 1/$f$ exponent ($rho = -0.94$, $p < 0.0001$; Fig 1B), while the overall amplitude had no signifi-cant impact with moderate evidence as suggested by the Bayes factor ($rho = -0.03$, $p = 0.8254$, $BF_{10} = 0.182$). Collectively, these two simulations indicate that timescales correlated with underlying periodic and aperiodic signal features, but did not enable an assessment of relative contributions in signals that mimic empirical timeseries.

Hence, we next simulated timeseries that were more physiologically plausible and exhibited both, 1/$f$ scaling char-acteristics as well as sinusoidal oscillations (Fig 1C). From these timeseries, we estimated both periodic neural activity (oscillatory peak above the 1/$f$ decay function) and the aperiodic timescale. We simulated 1,000 timeseries with varying physiologically plausible exponents (ranging from 1 to 4) and oscillations (ranging from 1 to 20 Hz). We predicted that both should be correlated if periodic and aperiodic temporal structure provide redundant information. However, we observed no statistically significant association between the peak frequency and the timescale, as assessed using Spearman's rank correlation and supported by the Bayes factor ($rho = 0.01$, $p = 0.8305$, $BF_{10} = 0.045$). However, we found a moder-ate correlation between oscillatory peak amplitude and the timescale ($rho = -0.12$, $p = 0.0001$). Critically, we observed a pronounced relationship between 1/$f$ aperiodic activity and the timescale ($rho = -0.46$, $p < 0.0001$). We bootstrapped and subsampled correlation coefficients (100 iterations) to obtain reliable estimates. Using repeated-measures ANOVA, we observed a significant difference between the correlations of the three parameters ($F = 406.74$, $p < 0.0001$). Post-hoc tests reveal that all correlations significantly differ ($p < 0.0001$). To address if timescales and peak amplitude overlapped in their variance explained, we employed a Spearman partial correlation. We observed that the correlation between the timescale and 1/$f$ noise remained unchanged after having accounted for oscillatory amplitude ($rho_{partial} = -0.47$, $p < 0.0001$), indicating no significant overlap in variance. In sum, these results demonstrate that the timescale mainly correlates with aperiodic signal features and to a much smaller degree with oscillatory peak amplitude, while no impact of oscillatory peak fre-quency was observed.

Lastly, we addressed a theoretical question arising from these considerations, namely if a timeseries that contained no periodic activity still exhibits aperiodic temporal regularities. Therefore, we employed a random walk model to simulate a timeseries characterized by a flat power spectrum (without oscillatory peaks), but maintained its autocorrelation structure; hence, exhibited an aperiodic timescale (Fig 1D). This scenario indicates that quantification of aperiodic timescales can capture underlying signal regularities that cannot be obtained from spectrally decomposed time series data.

In sum, these simulations indicate that periodic and aperiodic temporal structure may provide complementary informa-tion in simulations.

## Aperiodic temporal structure governs human spatial attention

After having addressed the relative contributions of periodic and aperiodic activity to timescales in simulations, we tested if both can be disentangled experimentally in behavior. We first assessed reaction times (RTs) as a function of a varying cue-target-interval in two tasks previously elicited rhythmic sampling behavior [9,10]. In Experiment 1a, 14 healthy partici-pants performed a variant of the classic Posner task where they had to monitor two spatial locations [25,26]. The same set of participants performed another variant of the task where they had to monitor four different locations [6,27] (Experiment 1b; $N = 14$). In both tasks, participants were cued to covertly attend to one of two or four locations and respond to a target after a variable cue-target interval, which could appear in the cued or non-cued location (Fig 2A). We assessed RTs as a function of the cue-target interval (Fig 2B; left), obtained spectral representations (Fig 2B middle) and extracted the decay time constant $\tau$ from the autocorrelation function (Fig 2B right).

We correlated the spectral power per frequency with the individual timescales to test if periodic and aperiodic temporal structures provide redundant or separate information (Fig 2C). We observed no significant correlation between spectral estimates and the aperiodic timescale in either task (Fig 2D; cluster-based correlations; no clusters at $p < 0.05$). Next,

we assessed the correlation between individual peak frequencies and aperiodic timescales, but again found no significant association, with weak to moderate evidence in favor of the null hypothesis (two locations: $rho = -0.27$, $p = 0.3597$, $BF_{10} = 0.48$; four locations: $rho = 0.03$, $p = 0.9308$, $BF_{10} = 0.33$).

In sum, this set of analysis demonstrates that aperiodic timescales capture temporal regularities, irrespective of the presence of rhythmic temporal structure, and can be reliably estimated from behavior.

## Aperiodic temporal structure indexes behavioral demands

Having established that aperiodic timescales provide complementary insights into the temporal structure of attention, we next tested their functional relevance. In the behavioral traces, we observed that aperiodic timescales were shorter, manifested by a quicker decay rate of the autocorrelation function, when only two locations were covertly monitored

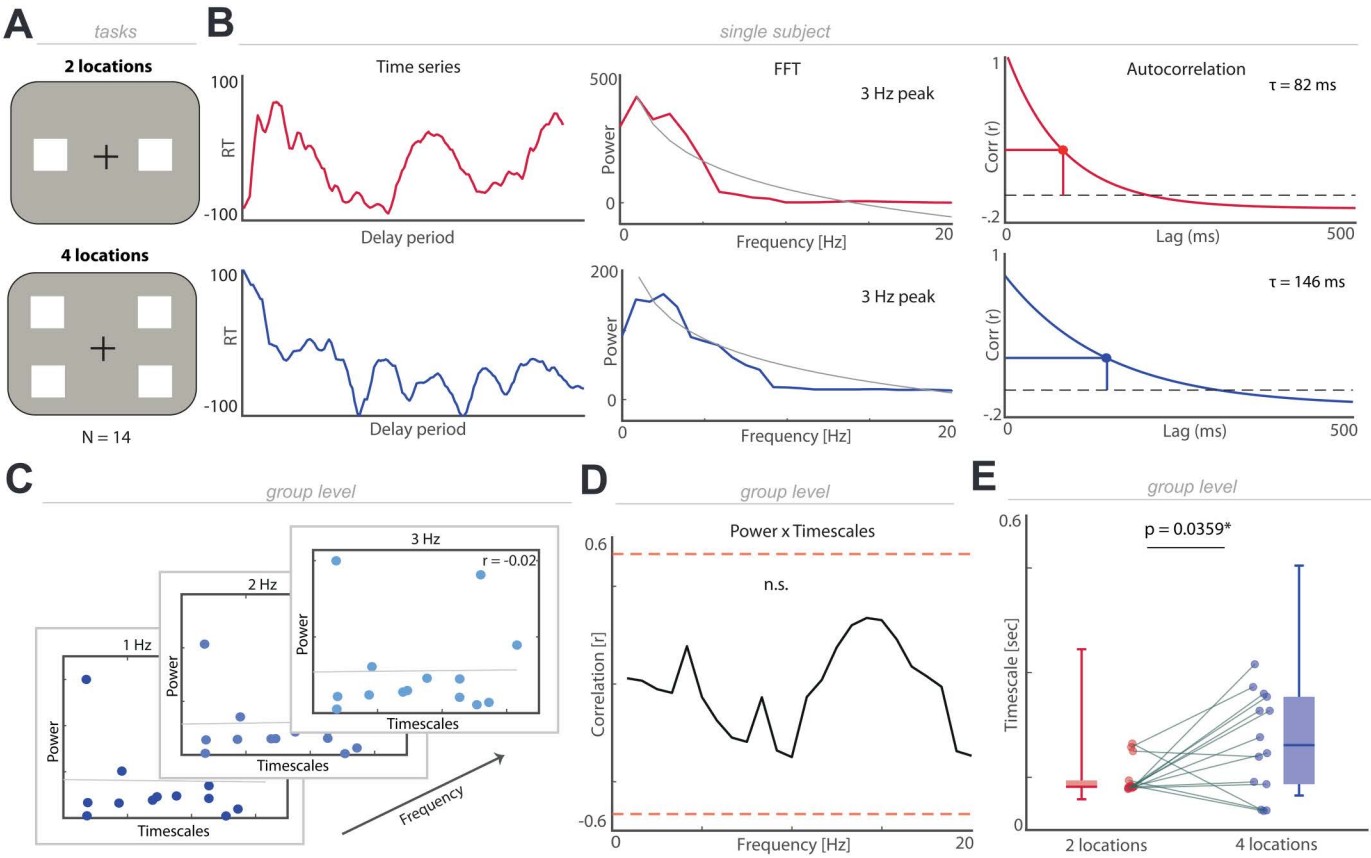

**Fig 2. Behavioral timescales increase when multiple locations are sampled. (A)** Schematic of task designs. Participants fixated a central cross and were presented with a cue, which indicated the location participants should covertly attend to. After a variable cue-target interval a target appeared in either the cued or non-cued location and participants responded with a button press. In the first task participants only had to sample two locations, while in the second task, participants had to sample four locations. **(B)** Left: demeaned, time-resolved RTs as a function of the cue-target interval for one exemplary participant (two locations: red; four locations blue). Middle: power spectrums with different peak frequencies. Right: the autocorrelation function and the respective timescales. **(C)** Correlation between timescales and behavior power per frequency (four location task). **(D)** Behavior power did not correlate with behavioral timescales (no cluster identified; red lines represent significance threshold; four location task; comparable results for two location task). **(E)** Timescales are longer as well as more variable when multiple locations need to be sampled in comparison to only two locations ($p = 0.0359$). The individual values for panel E are included in S2 Data, Experiment 1 sheet.

(0.09 ± 0.01 s; mean ± SEM) as compared to when four locations were sampled (Fig 2E; 0.16 ± 0.03 s; mean ± SEM; $t_{13} = -2.34$, $p = 0.0359$, $d = -0.93$; two-tailed $t$ test).

We conducted several control analyses to assess whether aperiodic timescales can be estimated from behavioral data. First, we repeated the timescale estimation after shuffling the association between the cue-target-interval and RTs (1,000 iterations). We observed that the shuffled autocorrelation functions approached zero more rapidly than the empirically observed timescale (S1A Fig, left; two locations: 0.08 ± 0.0005 s; mean ± SEM; [0.0836,0.0843]; 95% CI; four locations: 0.05 ± 0.001 s; mean ± SEM; [0.0494,0.0514]; 95% CI); indicating that observed timescales reflect true autocorrelations above chance level (two locations: $p < 0.0001$; four locations: $p < 0.0001$; z-scored relative to shuffled distribution).

We repeated the analyses after shifting the temporal window (50 different random starting points) that was used for the estimation of the autocorrelation function to test the stability of aperiodic timescales were stable or whether they were longer at the start of the task (e.g., due to a hazard function) (S1A Fig, right). By correlating the mean of the shifted timescales and the original values, we observed highly similar estimates (two locations: $rho = 0.67$, $p = 0.0089$; four locations: $rho = 0.78$, $p = 0.0009$), indicating that aperiodic timescales are robust over time.

We re-computed the aperiodic timescales using different window sizes and smoothing windows to examine the effect of averaging window size and smoothing (S1 Fig, Materials and methods). These findings indicate that different analytical parameters have a negligible impact of under 0.03 s on timescale estimation. We employed the same definitions throughout all analyses to provide consistent estimates.

## Neural timescales predict behavior during spatial attention

To determine how neural timescales influence behavior, we recruited a different group of participants who performed the variant of the attention task where two spatial locations were monitored ($N = 23$; experiment 2), while simultaneous whole-head 64-channel EEG was acquired [10].

We observed that behavioral timescales did not differ between both cohorts with moderate evidence in favor of the null hypothesis (experiments 1a and 2; Fig 3A; $t_{35} = 0.19$, $p = 0.8501$, $d = 0.06$, $BF_{10} = 0.33$; two-tailed $t$ test). Having

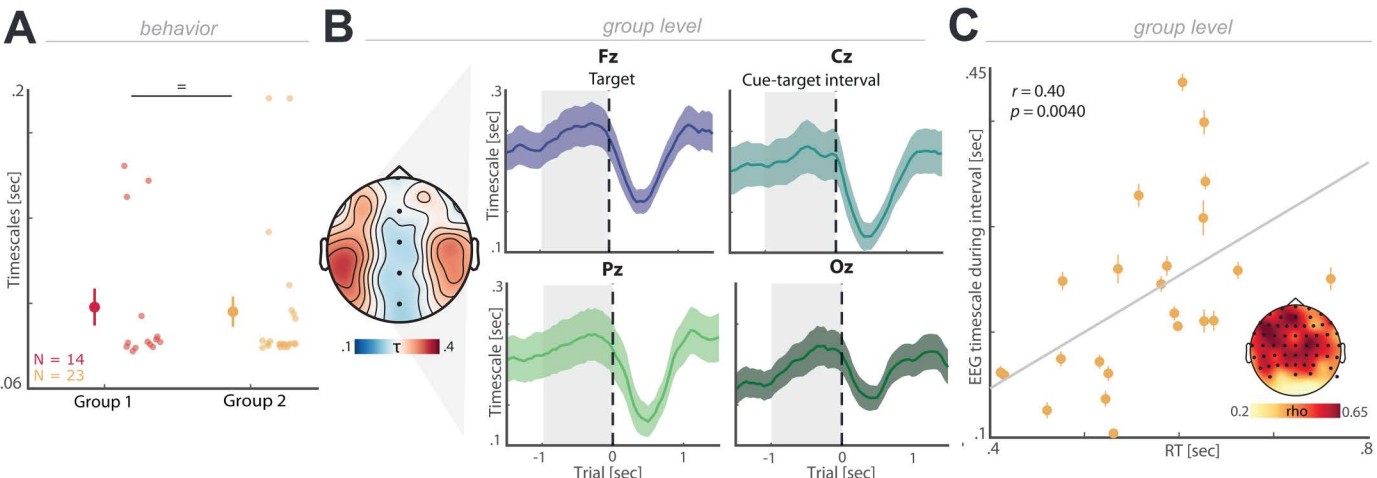

**Fig 3. Neural timescales predict behavior. (A)** Behavioral timescales do not differ between different groups of participants doing the same task (two locations) ($p = 0.8501$). **(B)** Grand-averaged target-locked neural timescales (mean ± SEM; shaded gray area represents cue-target interval) throughout the trial for four example channels. Topography of timescales during cue-target interval averaged across all subjects, channels Fz, Cz, Pz, and Oz highlighted. **(C)** Neuronal timescales correlate with reaction times (averaged target-locked timescales across the cue-target interval (from −1 to 0 s), error bars correspond to channels; the gray line highlights the linear regression; $p_{cluster} = 0.0040$). Inset: topography indicates spatial extent of the effect (black dots indicate cluster electrodes). The individual values for panel A are included in S2 Data, Experiment 1 and 2 sheets.

demonstrated that behavioral timescales are consistent across different groups, we next estimated aperiodic timescales from the EEG signals (termed neural timescales). Time-resolved neural timescales were obtained per trial and channel from the autocorrelation functions time-locked related to target- and cue-locked presentations using a 500 ms time window (Figs 3B and S2A). A decrease in timescales, in comparison to the cue-target interval, was observed after the target appears (0–500 ms) (Fz: $0.19 \pm 0.02$ s; mean $\pm$ SEM; $t_{22} = -7.29$, $p < 0.0001$, $d = -0.52$; Cz: $0.16 \pm 0.02$ s; mean $\pm$ SEM; $t_{22} = -5.88$, $p < 0.0001$, $d = -0.47$; Pz: $0.17 \pm 0.03$ s; mean $\pm$ SEM; $t_{22} = -5.67$, $p < 0.0001$, $d = -0.41$; Oz: $0.18 \pm 0.02$ s; mean $\pm$ SEM; $t_{22} = -5.86$, $p < 0.0001$, $d = -0.32$; two-tailed $t$ test), as the results of visual stimulus onset or the overt motor response.

To determine whether neural timescales predict behavior, we tested if neural timescales during the cue-target interval prior to target presentation predicted subsequent RTs across participants. A cluster-based correlation test revealed a positive association (Fig 3C; rho $= 0.40$, $p_{cluster} = 0.0040$), which indicated that shorter neural timescales, particularly over fronto-parietal channels, predict faster behavioral responses. To assess whether behaviorally relevant neural timescales were frequency-specific, we performed the timescale estimation as a function of power in every frequency band and observed that the theta-band timescale (~4 Hz) approximated the numerical values of behavioral timescale (S2B Fig; $0.08 \pm 0.001$ s; mean $\pm$ SD). In sum, these results demonstrate that neural timescales do not reflect a static property, but are dynamic and predictive of behavioral performance.

## Neural timescales index behavioral demands during attentional allocation

After having established group-level functional relevance, we addressed whether timescale dynamics index behavioral demands within participants across different behavioral contexts. We leveraged intracranial EEG recordings in humans with enhanced superior spatiotemporal resolution enabling linking single trial dynamics to behavioral outcome [28]. We recorded intracranial EEG in patient cohorts that were implanted with subdural grid electrodes that covered the frontoparietal attention network. The first group ($N = 8$; experiment 3) performed the two-location spatial attention task (cf. experiment 1a/2), while the second group ($N = 7$; experiment 4) performed the four-location spatial attention task (cf. experiment 1b).

We replicated the previous behavioral finding (Fig 3A) and found no difference between behavioral timescales of different groups of participants in both the two and four-location task, with weak to moderate evidence according to the Bayes factor (Fig 4A top; two locations: $t_{20} = 0.01$, $p = 0.9934$, $d = 0.004$, $BF_{10} = 0.39$; four locations: $t_{19} = 0.39$, $p = 0.6972$, $d = 0.18$, $BF_{10} = 0.43$; two-tailed $t$ test). As a control, we also tested for differences between timescales for targets appearing in the left or right hemifield and did not observe differences between the left and right hemifield timescales, even though the evidence is weak to moderate as suggested by the Bayes factor analysis (two locations: $t_7 = -0.61$, $p = 0.5585$, $d = -0.32$, $BF_{10} = 0.39$; four locations: $t_6 = 0.88$, $p = 0.4102$, $d = 0.38$, $BF_{10} = 0.48$; two-tailed $t$ test).

Both patient groups had an extensive frontal, motor, and parietal coverage (Fig 4A, bottom). Given the number of participants, we analyzed the intracranial data at the pseudo-population level by pooling the electrodes together, analogous to neurons in non-human primate experiments [16,21]. We determined the time-resolved neural timescales per channel and time window and observed a pattern that resembled our scalp level observations (Fig 4B; cf. Figs 3 and S3). Again, we observed that after the target appears (0–500m), sensory input as well as response execution decreased the neural timescale, in comparison to the cue-target interval, (Frontal: $0.19 \pm 0.008$ s; mean $\pm$ SEM; $t_{180} = -4.10$, $p < 0.0001$, $d = -0.06$; Motor: $0.33 \pm 0.029$ s; mean $\pm$ SEM; $t_{59} = -4.58$, $p < 0.0001$, $d = -0.12$; Parietal: $0.29 \pm 0.012$ s; mean $\pm$ SEM; $t_{272} = -8.16$, $p < 0.0001$, $d = -0.08$; two-tailed $t$ test).

Having established that timescales dynamically change over the course of a single trial, we next examined whether neural timescales also indexed behavioral demands during attention allocation. We first determined spatially selective channels that exhibited a selective high-frequency band (HFB) activity, a proxy for population spiking activity in humans [29,30], (HFB; 70−150 Hz; thresholded at $z = 1.96$, corresponding to a two-tailed $p < 0.05$; time window: 50 ms) increase

during cue presentation on trials that were presented in the visual field contralateral to the intracranial EEG grid electrodes (Fig 4C, top). This analysis approach is comparable to responses within and outside of a receptive field [24]; albeit, at the electrode and not the single neuron level. We then compared the neural timescales during trials when attention was allocated to the contralateral (attend-in; relative to the electrode grid) visual field versus trials when attention was shifted to the ipsilateral visual field (attend-out). We hypothesized that attend-in timescales would be shorter than attend-out timescales, as we previously observed that shorter attentional timescales are beneficial (cf. Fig 3C). We observed that during both variants of the attention task, the two- and four-location task, timescales prior to target presentation (1 s) decreased for attend-in in comparison to attend-out condition (two locations: $0.23 \pm 0.01$; mean $\pm$ SEM; $t_{738} = -2.25$, $p = 0.0246$, $d = -0.17$; four locations: $0.24 \pm 0.01$; mean $\pm$ SEM; $t_{661} = -4.02$, $p < 0.0001$, $d = -0.31$; two-tailed $t$ test).

In sum, the results indicate that neural timescales are dynamic in nature and reflect the current behavioral demands. Shorter timescales were observed during attentional allocation in accord with the behavioral cueing benefits (cf. Fig 3C).

## Neural timescales index non-spatial attentional demands and predict behavior

Finally, we addressed whether the observed attentional timescale modulations are specific to spatial attention tasks or whether they scale with cognitive demands and engagement. We reasoned that neural timescales might become longer if information has to be maintained over longer periods of the time, such as during working memory operations.

To test this hypothesis, we recorded intracranial EEG from PFC while participants performed a visual search task ($N = 8$; experiment 5). Note, here we focused on frontal channels given that recordings were obtained from stereotactically placed depth electrodes (sEEG) and not grid arrays (ECoG), which did not provide a sufficient coverage over motor and parietal regions. In this task, participants were asked to encode a target stimulus (colored triangle of a certain orientation) and then report its location in a search display. The target color either differed (pop-out) or the same (search) as

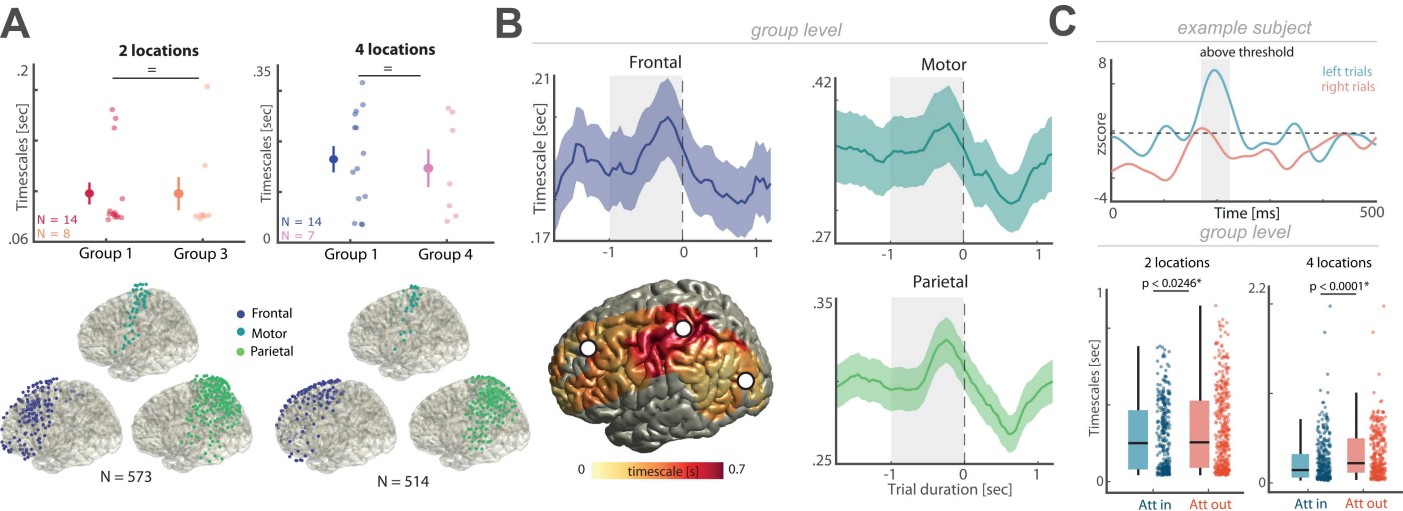

**Fig 4. ECoG timescales decrease during spatial attention. (A)** Top: behavioral timescales do not differ between different groups of participants (exp. 1 and 3/4) doing the same task; Bottom: Illustration of ECoG electrode placement for all subjects for both two location (left) and four location (right) tasks for the frontal, motor and parietal regions. **(B)** Grand-averaged target-locked neural timescales (mean $\pm$ SEM; shaded gray area represents cue-target interval) for three ROIs. Bottom left: Topography of timescales during the cue-target interval (white dots indicate the ROIs). **(C)** Top: example of selective channel for left hemifield-cued trials to illustrate analysis approach to identify spatially selective channels. If power was above the threshold ($z > 1.96$) for a minimum of 10% of the time window (500 ms) after the cue, the channel was considered hemifield-selective. Bottom: timescales significantly decrease during covert attention relative to the attend-out condition (two locations: $p = 0.0244$; four locations: $p < 0.0001$; mean $\pm$ SEM; whiskers indicate maximum and minimum; dots correspond to individual electrodes). The individual values for panel A and C are included in S2 Data, Experiment 1, 3 and 4 sheets and S3 Data, respectively.

the distractors, giving rise to a behavioral advantage in the pop-out condition (Fig 5A). We observed faster RTs during the pop-out condition in comparison to the search condition (Fig 5B; Pop-out: 708.44 ± 13.27 ms, mean ± SEM; Search: 960.18 ± 16.63 ms, mean ± SEM; $t_{714} = −11.90$, $p < 0.0001$, $d = 0.89$; two-tailed $t$ test).

We hypothesized that the behavioral pop-out effect might be indexed at the neural level by shorter neural timescales in the pop-out than the search condition since participants had to maintain the relevant target stimulus for longer in the search condition (Fig 5C). Response-locked neural timescales were extracted from a total of 220 frontal channels for both conditions (Fig 5D). To identify timescales after the stimulus onset, but before the motor response, we averaged the neural timescales in a 500 ms window before the response. In line with our hypotheses, we observed longer timescales in the search condition in comparison to the pop-out condition (Fig 5D right bottom; $t_{220} = −6.99$, $p < 0.0001$, $d = −0.25$; two-tailed $t$ test). To control for the temporal specificity, we performed the same analysis over time, which indicated that timescales only started to diverge upon onset of the search display (−0.5 s). As a control region, we applied the same analysis to channels in the lateral and ventral temporal lobe, where no statistically significant difference was observed with moderate evidence in favor of the null hypothesis ($t_{29} = −1,22$, $p = 0.2310$, $d = −0.17$, $BF_{10} = 0.38$; two-tailed $t$ test). Hence, these results suggest that the dynamic modulation of the neural timescale were spatially and temporally specific.

## Discussion

The findings across five experiments demonstrate that the neural basis of human attention-guided visual perception is also governed by aperiodic temporal regularities that structure the allocation of spatial and non-spatial attention. In addition to periodic regularities [4,6,9], we differentiate aperiodic temporal regularities using a multimodal convergent approach with simulations as well as behavioral and neural data. Our results reveal that aperiodic timescales are dynamically modulated in a spatially-, context- and demand-dependent manner. Overall, neural timescales increased from sensory to association cortex, decreased during sensory processing or action execution and were prolonged with increasing behavioral

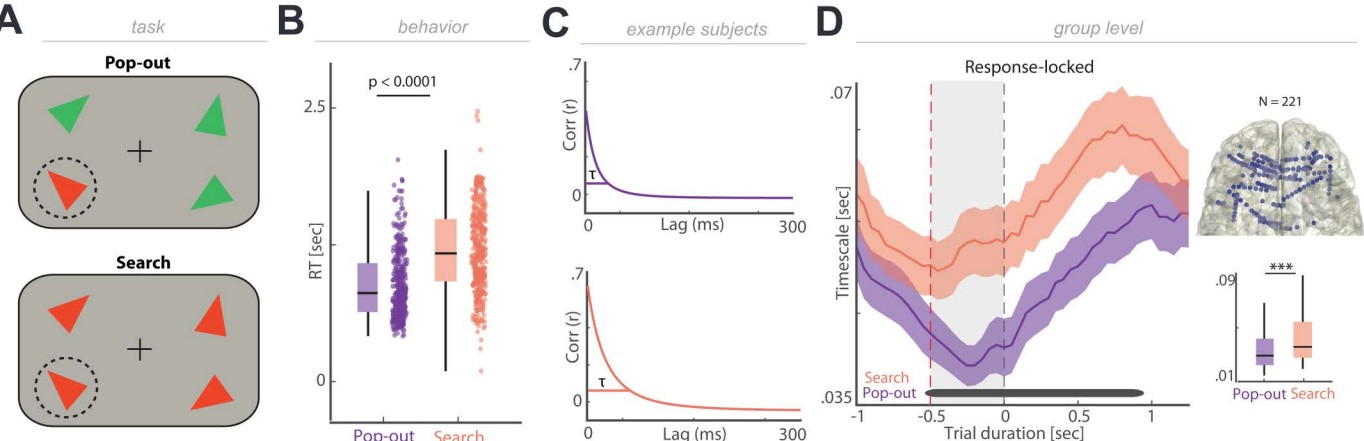

**Fig 5. Neural timescales increase with cognitive demands. (A)** Schematic task design. Patients were instructed to search for a target triangle ("sample", highlighted by dashed circle) of a given color and orientation. Once they identified the target triangle, they responded with a button press indicating whether its location was on the left or right half of the screen. **(B)** Reaction times during pop-out condition are shorter than during visual search condition ($p < 0.0001$). Individual dots correspond to trials. **(C)** Example of autocorrelation functions for each condition calculated on response-locked data. We hypothesized pop-out condition would show shorter timescales than the search condition. **(D)** Left: Grand-averaged response-locked neural timescales across frontal channels. Timescales in the period following search display onset (red dashed line) and after button press (black dashed line) decrease in pop-out condition relative to visual search condition returning to baseline around 1s (black bar represents significant time points $p < 0.05$). Right top: Topography of iEEG electrode placement in frontal ROI. Right bottom: Averaged timescales in the 500 ms period between search display onset and button press (gray shaded area) significantly differ between the conditions ($p < 0.0001$). The individual values for panel B and D are included in S2 Data, Experiment 5 sheet and S4 Data, respectively.

demands. In sum, these results provide a complementary perspective to contemporary theories highlighting the importance of periodic brain activity for attentional allocation and reconcile disparate previous findings.

## The temporal structure of attention

Spatial attention has traditionally been conceptualized as a 'static spotlight' that enhances sensory representations at the cued location [3] mediating behavioral benefits, such as faster RTs or higher detection accuracies. The recent observation that attention-guided visual perception fluctuates over time as a function of the phase of underlying rhythmic brain activity gave rise to a rhythmic theory of attention, which conceptualizes attention as a moving spotlight that sequentially samples task-relevant spatial locations [6,9,31,32]. Converging evidence across behavioral, electrophysiological and perturbation experiments in humans and non-human primates indicates that theta oscillations (~3–8 Hz) in the frontoparietal network support the interplay between covert sensory sampling and overt exploration of the visual environment [2,4,7,10,33,34]. A central challenge to this proposal is the quantification of rhythmic activity in time-resolved behavior or neural recordings. Several complementary approaches were introduced over the last decade that aim to establish the presence of oscillatory activity, such as irregular resampling, spectral parametrization or permutation statistics [35–37]. Recently, it was argued that commonly employed analysis tools might not test for rhythmicity per se, but rather test for the presence of any underlying temporal structure [12]. Based on autoregressive modeling, it had been suggested that aperiodic temporal regularities might shape the time course of attentional allocation. However, it remains unclear if periodic and aperiodic activity can be disentangled during attentional processing. Here, we estimated the aperiodic temporal structure from the decay of the autocorrelation function—a method that is commonly used to estimate 'intrinsic neural timescales' [19,38]. Our results reveal that aperiodic temporal structure constitutes an inherent feature of behavioral and neural recordings during attention-guided visual perception, providing non-redundant complementary information to periodic activity. Specifically, we did not find any evidence in simulations or experimental data that oscillations and the timescale were correlated. However, we observed that shuffling the underlying trial structure significantly reduced the behavioral timescale by dissipating the underlying aperiodic temporal structure [12]. Future work involving biologically plausible neural network simulations that generate both, periodic and aperiodic components of activity, reflecting signal generation in the brain more closely, could help provide a deeper understanding about these components are their interaction. An additional important future direction would be to address other forms of oscillatory phenomena such as high-frequency bursts, which have been shown to support attentional performance and facilitate attentional allocation [39]. Furthermore, while our results provide no evidence for a systematic nonlinear relationship between periodic and aperiodic temporal structures, future research could further investigate potential dependencies using alternative computational approaches. In sum, our results posit that aperiodic temporal structure is an inherent often neglected phenomenon governing visual attention and should be taken into account when assessing temporal regularities in behavior.

## Attentional modulation of timescales

Multiple lines of research indicate that neural timescales constitute a macroscopic principle of hierarchically structured neural activity. Single neuron recordings in animal models, scalp and intracranial EEG or fMRI in humans, demonstrate that neural timescales are shorter in sensory areas and longer in in association cortex [19,22,23,40,41]. Moreover, neural timescales have been shown to change in different arousal states [42–44] and pathological conditions [45–47]. Previously, timescales were often conceptualized as static properties that reflect an inherent property of the underlying neuronal population and could potentially reflect microscale cellular properties, such as the decay constants of cellular receptors [17,48,49]. More recently, it became evident that neuronal timescales might support cognitive operations. For example, it had been shown that working memory maintenance prolongs neural timescales in PFC [16,17], i.e., in a region supporting short-term mnemonic representations. It has also been shown that spatial attention increases neural timescales (at the level of single neuron recordings in non-human primate area V4; i.e., in sensory cortex) during attentional deployment,

while disengagement increased neural timescales [24]. Here, at the level of local population activity as recorded by scalp and intracranial EEG, we observed that timescales become shorter during allocation of attention, and that shorter timescales predict faster RTs. While these observations seem to contradict the previous observations by Zeraati and colleagues [24], it is important to highlight that we primarily recorded from association and not sensory cortex. It is well established that timescales differ along the cortical hierarchy. In line with our current result, Gao and colleagues [17] demonstrated that timescales are longer during working memory maintenance in PFC, which directly resembled our findings in experiment 5 during visual search, when the template stimulus had to be kept active until the target was correctly identified. These considerations raise the question: why did timescales become shorter during attentional allocation in association cortex? A testable hypothesis for future studies is that shorter timescales reflect sampling windows with high fidelity at the cued location. In contrast, at the uncued location longer sampling windows might be necessary to mitigate effects of more diffuse sensory representation at the unattended location. An inherent limitation of the current task design is that participants were instructed to withhold responses to targets at the uncued location, hence, impeding a dedicated behavioral analyses for targets at the uncued location. In sum, these theoretical considerations are in line with recent proposals that timescales might reflect 'temporal integration windows' [19,40,41].

## The neurophysiology of neural timescales

In its original definition, the concept of 'intrinsic neural timescales' referred to the characteristic time constant over which individual neuronal populations exhibit correlated activity patterns. Currently, the physiology behind neural timescales is not well understood, but it had been suggested that distinct timescales might *either* reflect physical properties of the underlying neuronal population, such as membrane leaks, dendritic filtering, or synaptic currents [17,49–52], *or* reflect functional interactions within the underlying population [38,53,54].

Here, we observed that timescales are functionally dynamic and change with cognitive demands, i.e., shorten during attentional allocation (Fig 4) and prolong during target maintenance (Fig 5), even though other factors, such as decision-making processes, may have also contributed to these effects. These results are in agreement with several lines of recent research that collectively demonstrated that timescales are hierarchically structured with longer timescales in association areas (c.f., Figs 3 and 4) needed to support the integration of task-relevant information during cognitive processing, such as previously observed during attentional engagement [24] working memory maintenance [16,17,55] or decision-making [21,56–58]. This dynamic nature of neural timescales implies that they do not solely represent physical properties of the underlying population, but more likely also index functional interactions and connectivity within the population. These considerations are in accordance with the observation that timescales change as a function of the underlying arousal state and, e.g., are prolonged during drowsiness, sleep [44] or general anesthesia [42,43]. Importantly, our results align with previous findings [59] showing that broadband and gamma timescales are not correlated, suggesting that the observed timescale differences are not simply a byproduct of high-frequency activity but rather reflect complementary neural dynamics.

The present results demonstrate that timescales decrease upon sensory input or action execution, given that both events introduce a short-lasted evoked response, which will reduce the signals autocorrelation. Hence, in order to reliably link timescales to behavior, it is critical to estimate them during temporal windows when stationarity can be assumed, such as during the delay period and not during stimulus presentation. In sum, our results provide additional support for the notion that neural timescales provide the neurophysiological basis for temporal integration windows, over which mnemonic information can 'be kept online' and hence, predicts memory retention, or over which visual information can be integrated to guide subsequent decision-making and action execution.

Lastly, it should be highlighted that multiple timescales may simultaneously be present within a given population. Timescales in single unit recordings and EEG/iEEG have been observed across a wide-range of temporal scales between ~2–4 [22,24] and ~50–200 ms [16,17,19,21,24]. While the very short timescales might possibly reflect AMPA or GABA

receptor time constants, longer timescales may index functional interactions [60]. Our current findings focus on the longer timescale amenable for EEG, but similar observations have been made at the level of single neuron recordings as exemplified by the findings by Zeraati and colleagues [24], who showed that attention changes timescales in the range from ~100 to 150 ms but not between ~2 and 10 ms.

## Conclusions

Collectively, our results establish the presence of behaviorally relevant aperiodic neural timescales in human attention. These results provide evidence that attention rhythms are distinct and are not an artifact of aperiodic temporal activity, and that both these temporal regularities jointly structure attentional processing in the human brain.

## Materials and methods

### Participants

In this study we included five independent experiments, which had previously been reported, albeit under a different question [9,10,61,62]. In Experiment 1a/b, we included 14 healthy adults which performed two different version of spatial attention tasks (8 males; 24.86 ± 5.55 years; mean ± SD) that were recruited from the University of California, Berkeley. This data had previously been reported in Helfrich and colleagues [9]. In Experiment 2, we recruited 23 healthy participants (12 males; 61 ± 14 years; mean ± SD) from the University of California, Berkeley, this data was previously reported in Raposo and colleagues [10]. For experiment 3, we obtained intracranial recordings from eight patients (five males; 30.63 ± 13.22 years; mean ± SD) from the Children's Hospital in Oakland, CA, USA ($N=1$), Johns Hopkins Hospital in Baltimore, MD, USA ($N=1$), and Stanford Hospital, CA, USA ($N=6$). For experiment 4, we recruited seven patients (two males; 35.29 ± 12.42 years; mean ± SD) from the University of California, Irvine Medical Center, USA ($N=6$) and the California Pacific Medical Center (CPMC), San Francisco, USA ($N=1$). Data from experiments 3 and 4 had previously been reported in Helfrich and colleagues [9]. In Experiment 5, we included eight patients (three males; 34.47 ± 12.24 years; mean ± SD) from the University of California, Irvine Medical Center, USA. This data contained the subset of participants from Hahn and colleagues [62] that were acquired in the context of Slama and colleagues [61]. All the intracranial recordings were obtained from epilepsy patients who underwent pre-surgical monitoring with implanted grid and depth electrodes. Electrode placement was exclusively guided by clinical considerations. All procedures were approved by the Institutional Review Board as well as by the Committee for Protection of Human Subjects at the University of California, Berkeley (Protocol number: 2010-02-783) or the Regional Committee for Medical and Healthy Research Ethics and conducted in agreement with the Declaration of Helsinki. All participants provided written informed consent to participate in the studies.

### Behavioral tasks

In experiments 1a, 2, and 3, all participants performed a variant of a Posner spatial attention task where two spatial locations had to be monitored [3,10,25]. Participants were instructed to maintain fixation on a central cross and were then asked to covertly shift their attention to either the left or right hemifield based on a centrally presented cue (70% validity). Following a variable cue-target interval ranging from 1,000 to 2,000 ms, a target (blue square) appeared. Participants were required to respond as fast as possible to targets presented in the cued hemifield and withhold their response when targets appeared in the uncued hemifield.

For experiments 1b and 4, participants completed a variant of the Egly-Driver task [6,9,27] (four location task). Two bar objects were presented accompanied by a brief spatial cue in the periphery around one bar, indicating the location where the target had the highest likelihood of appearing (72% cue validity). Following a variable cue-target-interval ranging from 500 to 1,700 ms, a target appeared either in the cued or uncued locations. Participants were instructed to respond as quickly as possible upon detecting the target. In contrast to previous reports and to enable comparability to experiments 1a/2/3, we analyzed RTs as a function of the cue-target-interval.

Lastly, in experiment 5, participants performed a visual search task [61]. The stimuli consisted of acute triangles in red or green. Participants' task was to identify a target triangle ("sample") among four displayed triangles on the screen. The target triangle was characterized by both its color (red or green) an orientation. Participants indicated the location of the target triangle, whether on the left or right half of the screen, by pressing the corresponding button.

## Neural data acquisition

**EEG.** EEG data were collected using a 64-channel BioSemi ActiveTwo with active electrodes mounted on an elastic cap according to the International 10-20 System (BioSemi, Amsterdam, Netherlands), sampled at 1,024 Hz. The data were offline re-referenced to a common average, de-meaned and linearly detrended, high-pass filtered at 0.3 Hz and low-pass filtered at 70 Hz using finite impulse response filters. Line noise and harmonics were removed using a 60 Hz band-stop filter. The data were then visually inspected for artifacts.

**ECoG/sEEG.** Intracranial data was manually inspected by a neurologist to identify channels with epileptiform activity and artifacts. ECoG arrays were organized in either grids or strips. ECoG data from every experiment was detrended, demeaned, filtered, and common average re-referenced as described previously [9,26]. In eight patients, sEEG depth electrodes targeting subcortical structures were implanted. Intracranial data were acquired using a Nihon Kohden recording system (UC Irvine, CPMC and Children's Hospital, 128/256 channel, 1,000/5,000 Hz digitization rate), a Natus Medical Stellate Harmonie recording system (Johns Hopkins, 128 channel, 1,000 Hz digitization rate) or a Tucker Davis Technologies recording system (Stanford, 128 channel, 3,052 Hz digitization rate).

## Simulation

First, we simulated pure sine waves in the absence of 1/$f$ noise, in order to isolate the contributions of frequency and amplitude to estimated timescales. Peak frequency was varied from 0.5 to 20 Hz in steps of 0.5 Hz and amplitude was varied across 50 logarithmically spaced values between 0.005 and 0.02 to match the amplitude of the noise signal within plausible limits. Each simulated signal $y(t)$ was generated as:

$$y(t) = A \sin \left(2\pi f \frac{t}{T}\right)$$

where $t$ represents each time step in the signal, $A$ the amplitude, $f$ the frequency, and $T$ is the total signal length.

Second, we generated signals with varying aperiodic 1/$f$ components. These signals were created using a power-law noise model, where the spectral exponent was varied between −4 and −1 across 50 logarithmically spaced values. Each signal $y(t)$ was generated as:

$$y(t) = A \, \eta(t, \, \beta)$$

where $A$ is the amplitude, and $\eta(t, \, \beta)$ represents a noise signal with a 1/$f^\beta$ power spectrum, where $\beta$ is the exponent. This process produced "colored" noise with a specific amplitude.

Next, we simulated a variety of biologically plausible time series, which were designed to generate a periodic signal with a specific frequency and amplitude oscillation. We generated a time series that exhibited 1/$f$ characteristics and incorporated an oscillatory component, with exponents ranging from −1 to −4, peak frequency between 1 and 20 Hz, and amplitude from 0.005 to 0.02, which scale within biologically plausible ranges. To simulate the 1/$f$ noise, we created a noise vector with power-law characteristics and we added a sinusoidal oscillation to this 1/$f$ noise to introduce a peak frequency and amplitude component. Each generated signal was defined as:

$$y(t) = \eta(t, \, \beta) + A \sin \left(2\pi f \frac{t}{T}\right)$$

Last, we simulated an aperiodic signal resembling a random walk process with a specific timescale. We employed a Markov process with probabilistic transitions to either maintain the current signal value($p_{stay}$)or introduce a new random value. $p_{stay}$ was set based on the stability period $x$, using the relationship$p_{stay} = 1 - \frac{1}{x}$. Specifically, each value $y(i)$ in the signal was generated as:

$$y(i) = \begin{cases} y(i-1), & \text{with probability } 1 - \frac{1}{x} \\ randn, & \text{with probability } \frac{1}{x} \end{cases}$$

We used a high probability of staying at the current signal value, as higher values result in more autocorrelation in the generated signal. We generated the random walk signal by iterating through each point $i$ along the signal length and deciding whether to stay at the current signal value or introduce a new random value based on the probability determined beforehand. We then calculated the autocorrelation of the signal as well as the Fast Fourier Transform. We repeated this process 1,000 times to ensure stability and calculated the mean autocorrelation and spectrum across repetitions.

### Timescales estimation

**Behavior.** Consistent with previous studies, we defined intrinsic timescales ($\tau$) as the exponential decay time constant of the autocorrelation function [15,16]. Temporal autocorrelation was computed per subject from behavioral time-courses during the cue-target interval. To extract the behavioral time-course, we shifted a 50 ms window in steps of 1 ms from 1,000–2,000 ms for experiment 1 and 500–1700 ms for experiment 2 and re-calculated the RTs across all validly cued trials in the respective time window. To interpolate missing data points from the data that resulted from the limited temporal sampling, the traces were further smoothed with a 25-point boxcar moving average. We fitted an exponential decay function to the resulting autocorrelation time course during the cue-target interval (1 s for experiment 1 and 1.2 s for experiment 2) using nonlinear least squares.

**EEG.** In neural activity, we estimated timescales across the whole trial. Temporal autocorrelation was calculated using a moving window of 500 ms in 50 ms steps throughout the trial per channel and a lag of 500 time points. To estimate the intrinsic timescale, we fitted an exponential decay function with an offset to the empirically estimated autocorrelation function. The exponential decay was defined as:

$$R(k\Delta) = A \left[ \exp\left( -\frac{k\Delta}{\tau} \right) + B \right]$$

where $\tau$ corresponds to the intrinsic timescale (i.e., rate of decay), $A$ is the amplitude and $B$ is the offset parameter. The $k\Delta$ parameter refers to the relative time lag between time bins. This procedure created a trial by channel by time-bin matrix of timescales. These timescales reflect the intrinsic neural dynamics within single trials. We excluded sessions with autocorrelations 3 standard deviations above the median, dominated by noise or strong deflections, that could not be well described with a mixture of exponential decay functions (approximately 1% of the timescales extracted).

### Definition of spatially selective electrodes

**HFB analysis.** We extracted the HFB activity by band-pass filtering the raw time courses in 14 non-overlapping 5 Hz wide bins ranging from 80 to 150 Hz and applying a Hilbert transform to extract the instantaneous amplitude with the default settings as implemented in Fieldtrip (ft_freqanalysis). Then every trace was separately baseline corrected by means of a z-score relative to a bootstrapped baseline distribution prior to cue onset (−0.2 s to 0 s, 1,000 iterations) [63] and subsequently averaged. Note that this approach accounts for the 1/$f$ signal drop off in the HFB with increasing frequencies. Finally, we discarded the edges to avoid filter artifacts.

**Spatially selective electrodes.** We classified an electrode as spatially selective toward the left or right location when the average HFB response to the cue exceeded a z-score of 1.96 (corresponding to a two-tailed *p*-value of 0.05) for at least 10% of consecutive samples in the cue period (0–0.5 s). This was calculated for left and right cued trials separately. Channels were then divided into ipsi- and contralateral to the cued hemifield relative to the grid location, corresponding to attention-out and attention-in, respectively.

## Statistical testing

Throughout, we report single-subject data and highlight effects that generalize across the population. Unless stated otherwise, we employed two-tailed paired *t*-tests (e.g., Fig 2C) and repeated-measures ANOVA in MATLAB (version R2021a) to infer significance at the group level. Given the small samples for the intracranial experiments, we also inferred statistical significance at the level of the pseudo-population. Furthermore, we employed cluster-based correlation tests to correct for multiple comparisons as implemented in Fieldtrip (Monte Carlo method; 1000 iterations; 0.05 cluster alpha; 0.025 alpha; maxsum criterion [37]) based on Pearson correlation coefficient, which was subsequently transformed into a *t*-statistic (Figs 1E and 3C). Effect sizes were calculated using Cohen's d or the correlation coefficient rho. For tests that yielded non-significant results in the frequentist statistics, we conducted Bayesian analyses using JASP [64] (version 0.19.1.0) to assess the strength of evidence in favor of the null hypothesis. $BF_{10}$ were computed, where values less than 1 indicate support for the null hypothesis relative to the alternative [65].

## Supporting information

**S1 Fig. Control analyses of behavioral timescales. (A)** Left: Randomly shuffling the time series data abolishes the signal autocorrelation; hence, removes the aperiodic temporal structure as indexed by decreasing timescales ($p < 0.0001$; *z*-scored relative to shuffled distribution). Right: Estimating timescales for different starting points maintains aperiodic temporal structure ($p = 0.2106$; *z*-scored relative to shifted distribution). **(B)** Two location task example: We observed no statistically significant differences between different smoothing windows for the two location task, but for the four location task (two locations: $F3,39 = 1.35$, $p = 0.2727$; four locations: $F3,39 = 32.15$, $p < 0.0001$, repeated-measures ANOVA). **(C)** Window size had a statistically significant effect on the behavioral timescales for both tasks (two locations: $F3,39 = 5.52$, $p = 0.0029$; four locations: $F3,39 = 6.18$, $p = 0.0015$, repeated-measures ANOVA; two location task). The individual values for panel B and C are included in S5 Data and S1B and S1C Figs.
(EPS)

**S2 Fig. Cue-locked timescale timeseries in EEG data. (A)** Grand-averaged cue-locked neural timescales (mean ± SEM; shaded gray area represents cue-target interval) throughout the trial for four example channels. Topography of timescales during cue-target interval averaged across all subjects. Black dots highlight the location of the four example channels. **(B)** Timescales per frequency obtained by applying a Hilbert transform, error bars correspond to subjects (mean ± SEM; channel Fz). Theta range timescale (~4 Hz) approximates the value of the behavioral timescale (~0.08 ms).
(EPS)

**S3 Fig. Cue-locked timescales timeseries in EcoG data.** Grand-averaged cue-locked neural timescales (mean ± SEM; shaded gray area represents cue-target interval) for three ROIs. Bottom left: Topography of timescales during the cue-target interval (white dots indicate the ROIs).
(EPS)

**S1 Data. Source data for Fig 1C.**
(XLSX)

**S2 Data. Source data for Figs 2E, 3A, 4A and 5B.**
(XLSX)

**S3 Data. Source data for Fig 4C.**
(XLSX)

**S4 Data. Source data for Fig 5D.**
(XLSX)

**S5 Data. Source data for Fig S1.**
(XLSX)

## Author contributions

**Conceptualization:** Ian C. Fiebelkorn, Sabine Kastner, Robert T. Knight, Randolph F. Helfrich.

**Data curation:** Randolph F. Helfrich.

**Formal analysis:** Isabel Raposo.

**Funding acquisition:** Josef Parvizi, Sabine Kastner, Robert T. Knight, Randolph F. Helfrich.

**Investigation:** Jack J. Lin, Josef Parvizi.

**Methodology:** Randolph F. Helfrich.

**Resources:** Ian C. Fiebelkorn, Assaf Breska.

**Software:** Assaf Breska.

**Supervision:** Randolph F. Helfrich.

**Visualization:** Isabel Raposo.

**Writing – original draft:** Isabel Raposo.

**Writing – review & editing:** Ian C. Fiebelkorn, Jack J. Lin, Josef Parvizi, Sabine Kastner, Robert T. Knight, Assaf Breska, Randolph F. Helfrich.

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
