## [Editor Report · Decision Letter 0]

20 Nov 2024

Dear Dr Raposo, 

Thank you for submitting your manuscript entitled "The aperiodic temporal structure of human attention" for consideration as a Research Article by PLOS Biology.

Your manuscript has now been evaluated by the PLOS Biology editorial staff as well as by an academic editor with relevant expertise and I am writing to let you know that we would like to send your submission out for external peer review.

Once your full submission is complete, your paper will undergo a series of checks in preparation for peer review. After your manuscript has passed the checks it will be sent out for review. To provide the metadata for your submission, please Login to Editorial Manager (https://www.editorialmanager.com/pbiology) within two working days, i.e. by Nov 22 2024 11:59PM.

Kind regards,

Christian

Christian Schnell, PhD

Senior Editor

PLOS Biology

cschnell@plos.org

---

## [Editor Report · Decision Letter 1]

2 Dec 2024

Dear Dr Raposo,

Thank you for your patience while we discussed the revision of your transferred manuscript "The aperiodic temporal structure of human attention" with our Academic Editor. 

Based on these discussions, we agree that many of the concerns raised by the reviewers at Nature Human Behaviour have been addressed. However, before sending the revised manuscript back to the reviewers, we would like to ask you to address a few concerns, which we think are not fully addressed so far:

* Reviewer 1 wrote "Second, even if we accept the lack of a correlation as evidence for orthogonality – there are several other links that can be demonstrated that are not linear." We do not think that this sufficiently addressed in the current revision/rebuttal letter.

* Reviewer 1 also wrote "Second, it is unclear why the authors chose to use such a high probability of staying at the current signal value. This results in a signal that is very unlike beahvioral or physiological signals, also raising questions over the reported findings." We think that your response makes sense and that this is about a proof-of-concept rather than about biological plausibility. However, we also think that you could have done better explaining this. Currently, the reviewer may not fully understand your reasoning and hence don’t think their point has been addressed.

* Reviewer 2 suggested to conduct a neural network simulation that gives rise to both a periodic and aperiodic component of activity. The reviewer would probably not fully agree with how you address this point and would likely request you to simulate a scenario where both aspects (1/f and oscillations) are correlated. Is there maybe a middle ground here you can use your current simplified simulation and have some shared variance between the two parameters?

Given the extent of revision needed, we cannot make a decision about publication until we have seen the revised manuscript and your response to the reviewers' comments. Your revised manuscript will be sent for further evaluation by all or a subset of the reviewers.

**IMPORTANT - SUBMITTING YOUR REVISION**

*Re-submission Checklist*

*Published Peer Review*

*PLOS Data Policy*

*Blot and Gel Data Policy*

Sincerely,

Christian

Christian Schnell, PhD

Senior Editor

PLOS Biology

cschnell@plos.org

---

## [Decision Letter · Decision Letter 2]

7 May 2025

Dear Dr Raposo,

Thank you for your patience while we considered your revised manuscript "The aperiodic temporal structure of human attention" for publication as a Research Article at PLOS Biology. This revised version of your manuscript has been evaluated by the PLOS Biology editors, the Academic Editor and one new reviewer.

Based on the review and on our Academic Editor's assessment of your revision, we are likely to accept this manuscript for publication, provided you satisfactorily address the remaining points raised by the reviewer and the following data and other policy-related requests:

* We would like to suggest a different title to improve its accessibility for our broad audience: 

Human attention-guided visual perception is governed by rhythmic oscillations and aperiodic timescales 

* Please add the links to the funding agencies in the Financial Disclosure statement in the manuscript details.

* DATA POLICY:

Regardless of the method selected, please ensure that you provide the individual numerical values that underlie the summary data displayed in the following figure panels as they are essential for readers to assess your analysis and to reproduce it: 1C, 2E, 3A, 4AC, 5BD and S1BC.

* CODE POLICY

We expect to receive your revised manuscript within two weeks. 

*Published Peer Review History*

*Press*

Sincerely,

Christian

Christian Schnell, PhD

Senior Editor

cschnell@plos.org

PLOS Biology

Reviewer remarks:

Reviewer #1 (Daniel S. Kluger): Overall, Raposo and colleagues provide concise and satisfactory responses to the issues raised by both reviewers. I only have one remaining concern I would consider to be of minor importance:

I did not find the authors' response to Reviewer 1's comment regarding the interpretation of absent correlations particularly convincing. The absence of a correlation alone is by no means sufficient to claim distinct processes, and I do think the authors' approach to amend frequency statistics by Bayes factor analyses is a good way to go. However, I was surprised that the authors merely added Bayesian statistics without any further comment - at least a brief interpretation of BF values should be added in case (frequentist) statistical comparisons were found to be non-significant.

---

## [Editor Report · Decision Letter 3]

29 May 2025

Dear Dr Raposo,

Thank you for the submission of your revised Research Article "Human attention-guided visual perception is governed by rhythmic oscillations and aperiodic timescales" for publication in PLOS Biology. On behalf of my colleagues and the Academic Editor, Simon Hanslmayr, I am pleased to say that we can in principle accept your manuscript for publication, provided you address any remaining formatting and reporting issues. These will be detailed in an email you should receive within 2-3 business days from our colleagues in the journal operations team; no action is required from you until then. Please note that we will not be able to formally accept your manuscript and schedule it for publication until you have completed any requested changes.

PRESS

Sincerely, 

Christian

Christian Schnell, PhD

Senior Editor

PLOS Biology

cschnell@plos.org